# Overcoming Immune Checkpoint Therapy Resistance with SHP2 Inhibition in Cancer and Immune Cells: A Review of the Literature and Novel Combinatorial Approaches

**DOI:** 10.3390/cancers15225384

**Published:** 2023-11-13

**Authors:** Alireza Tojjari, Anwaar Saeed, Arezoo Sadeghipour, Razelle Kurzrock, Ludimila Cavalcante

**Affiliations:** 1UPMC Hillman Cancer Center, University of Pittsburgh, Pittsburgh, PA 15232, USA; 2Department of Biochemistry, Faculty of Biological Sciences, Tarbiat Modarres University, Tehran P.O. Box 14115-175, Iran; 3Department of Medicine, Genome Sciences and Precision Medicine Center, Medical College of Wisconsin Cancer Center, Milwaukee, WI 53226, USA; 4Novant Health Cancer Institute, Charlotte, NC 28204, USA

**Keywords:** SHP2, cancer, T-cell, PD-L1, immunotherapy

## Abstract

**Simple Summary:**

This is a thorough literature review of SHP2 inhibition, its application in cancer therapy and current uses as an immune modulator. The aberrant activation of SHP2, a protein tyrosine phosphatase widely expressed in many cell types, is implicated in multiple human cancers. As SHP2 is at the crossroads of many functions and pathways, this molecule can be leveraged for therapeutic targeting in cancer. SHP2 inhibitors have shown promise in overcoming resistance to kinase inhibitors and PD-1 blockade by hindering the activation of compensatory signaling pathways. We also explore possible combinatorial approaches, wherein SHP2 inhibitors are used to overcome drug resistance, especially to immunotherapy agents. Many SHP2 inhibitors have emerged, and several have reached clinical development in solid tumors. The field is now evolving into combinatorial approaches and new delivery methods, such as SHP2 degraders.

**Abstract:**

SHP2 (Src Homology 2 Domain-Containing Phosphatase 2) is a protein tyrosine phosphatase widely expressed in various cell types. SHP2 plays a crucial role in different cellular processes, such as cell proliferation, differentiation, and survival. Aberrant activation of SHP2 has been implicated in multiple human cancers and is considered a promising therapeutic target for treating these malignancies. The PTPN11 gene and functions encode SHP2 as a critical signal transduction regulator that interacts with key signaling molecules in both the RAS/ERK and PD-1/PD-L1 pathways; SHP2 is also implicated in T-cell signaling. SHP2 may be inhibited by molecules that cause allosteric (bind to sites other than the active site and attenuate activation) or orthosteric (bind to the active site and stop activation) inhibition or via potent SHP2 degraders. These inhibitors have anti-proliferative effects in cancer cells and suppress tumor growth in preclinical models. In addition, several SHP2 inhibitors are currently in clinical trials for cancer treatment. This review aims to provide an overview of the current research on SHP2 inhibitors, including their mechanism of action, structure-activity relationships, and clinical development, focusing on immune modulation effects and novel therapeutic strategies in the immune-oncology field.

## 1. Introduction

The modular role of tiny protein domains has emerged as a new subject in signal transduction biology and protein biochemistry. These can, in certain situations, directly control catalytic activity. In others, they function to bind essential regulatory proteins. Some of the most well-studied models are SH2 (src homology 2) domains [1]. In 1994, scientists discovered a phosphorylated protein linked to the protein Shc1 after BCR or cytokine receptor activation [2]. It has two significant isoforms: SHP1 and SHP2 [3]. SHP2 interacts with its targets via its SH2 domains, and binding via both SH2 domains results in its maximal enzymatic function [4].

SHP2 is also a well-known modulator of multiple other signaling pathways, making it an appealing anti-cancer therapeutic target [5]. Various cancer cell signaling pathways that promote cancer development, such as receptor tyrosine kinases (RTKs), MAPK, PI3K-AKT-mTOR, and JAK-STAT, play a role in making immune checkpoint therapy less effective. These pathways involve the participation of SHP2 [6]. Moreover, immune checkpoint therapy resistance also arises from immune-suppressive signaling pathways like PD-1 and CTLA-4 signaling, in which SHP2 is implicated [7]. SHP2 influences the tumor microenvironment (TME) by promoting immunosuppressive factors, including cytokines and chemokines, which create an unfavorable immune response environment. Additionally, SHP2 affects TGF-beta signaling, contributing to cancer progression through enhanced epithelial-mesenchymal transition (EMT) and immunosuppression in the TME [8]. SHP2’s impact extends to myeloid cells, where its interaction with the PD-1-SHP2 axis affects GM-CSF-mediated protein phosphorylation, influencing myeloid differentiation and potentially offering opportunities for cancer immunotherapy [9].

Recently, studies have demonstrated the therapeutic potential of SHP2 inhibitors in overcoming drug resistance to kinase inhibitors and programmed cell death protein 1 (PD-1) inhibition. The emergence of immune checkpoint inhibitors (ICIs) has significantly advanced cancer treatment, but resistance to ICIs remains a major challenge [10]. Inhibiting SHP2 can enhance the efficacy of ICIs by modulating the TME, bolstering immune surveillance, and promoting the elimination of cancer cells [11,12]. SHP2 also regulates key signaling pathways involved in antigen-presenting cell (APC) activation and antigen presentation (such as PI3K-AKT and JAK-STAT). Inhibition of SHP2 enhances antigen presentation by APCs, leading to increased recognition of tumor antigens by T cells and subsequent immune-mediated tumor cell killing [12,13]. SHP2 inhibitors have shown promise in overcoming resistance to kinase inhibitors and PD-1 blockade by hindering the activation of compensatory signaling pathways [14]. However, the development of potent and selective SHP2 inhibitors with favorable pharmacokinetic properties remains a challenge [15]. Additionally, understanding the complex interactions between SHP2 and other signaling molecules within the TME is crucial for optimizing therapeutic strategies [16].

## 2. The Structure of SHP2

SHP2 consists of two SH2 domains (called N-SH2 and C-SH2), a protein tyrosine phosphatase domain with catalytic activity (PTP domain), a C-terminal tail with two tyrosine phosphorylation sites (Y542 and Y580), and a proline-rich motif (Figure 1A) [17].

The function of SHP2 is regulated by its structural conformation between closed (or basal) and opened states (Figure 1B) [18]. When SHP2 is in its closed position, auto-inhibition happens because the back of the N-SH2 domain forms a loop and interacts intramolecularly with PTP, which has a catalytic pocket. This interaction blocks the catalytic pocket and stops SHP2’s substrates from getting to it (Figure 1C) [17,18].

When growth factors and cytokines stimulate cells, SH2 domains bind to phosphorylated tyrosine residues on various proteins, including cytokine receptors, receptor tyrosine kinases, and scaffolding adaptors. Once a phosphotyrosyl peptide binds to the N-SH2 domain, SHP2 undergoes a conformational change, which exposes its catalytic pocket. This open, active conformation allows the SHP2 substrates to access the catalytic pocket and leads to dephosphorylation (Figure 1D) [18,19,20].

## 3. Immune Checkpoints

Immune checkpoints are a multitude of natural inhibitory pathways ingrained within the immune system. These pathways are vital in preserving self-tolerance and regulating the extent and duration of immune responses in peripheral tissues [21]. Different immune cells, including T cells, NK cells, macrophages, and dendritic cells, express various immune checkpoint receptors on their surface. Each of these immune cells has distinct ligand molecules specifically expressed in tumor cells [22]. Immune checkpoints, particularly CTLA-4, PD-1, and PD-L1, can negatively regulate T cells by activating SHP2, leading to immune tolerance and allowing tumors to evade immune surveillance [23,24,25].

## 4. Immune Checkpoint Therapy

Immune checkpoint therapy is a revolutionary approach to cancer treatment that harnesses the body’s immune system to target and eliminate cancer cells. It involves the use of monoclonal antibodies that block the inhibitory signals generated by immune checkpoints, thereby reinvigorating the antitumor immune response and unleashing the full potential of T cells to recognize and attack cancer cells. By releasing the brakes on the immune response, this therapy enhances the immune system’s ability to detect and destroy tumors [26]. While well-studied immune checkpoints like PD-1, CTLA-4, and PD-L1 have shown clinical benefits, an expanding range of other immune checkpoints are being investigated for their potential benefits [27].

## 5. Immune Checkpoint Therapy Resistance Mechanisms

Immune checkpoint therapy, while often effective, can face challenges due to the development of resistance mechanisms. Several factors contribute to immune checkpoint therapy resistance, including:

### 5.1. TME Changes

The alterations within TME can lead to immune checkpoint inhibition and resistance through various mechanisms. Immunosuppressive cytokines, regulatory T cells, myeloid-derived suppressor cells (MDSCs), exhausted T cells and molecules like indole-2,3-dioxygenase (IDO) contribute to an immunosuppressive environment. Factors intrinsic to tumor cells, including mutational load, oncogenic signaling pathways, PD-L1 expression, and MHC class 1 downregulation, also play a role. The elevated secretion of inhibitory cytokines such as IL-6, IL-10, and TGF-beta by tumor cells contributes to immune tolerance, hindering cytotoxic T-cell function and promoting Treg and MDSC infiltration. High expression of immune suppressive molecules like IDO and PD-L1 counters immune activation even when CD8-T cells are present. Tryptophan catabolism by IDO leads to immunosuppressive metabolites that inhibit T-cell expansion. PD-L1 expression by tumor cells also influences immune escape [28,29].

### 5.2. Loss of Antigen Expression

Cancer cells may downregulate the expression of antigens that T cells recognize. Any process that contributes to the reduction in the presentation and expression of tumor antigens can result in the development of acquired resistance to checkpoint inhibitors. Tumors that undergo immune editing and select low-immunogenicity sub-clones with reduced neo-antigen expression can become resistant to checkpoint inhibitors. Strategies to overcome the lack of tumor immunogenicity include adoptive cell therapy (ACT) with T-cells targeting specific antigens, creating mutational neo-antigens through vaccinations, and using epigenetic drugs to increase tumor antigen expression [30].

### 5.3. New Genetic Mutations

Gene mutations in both tumor cells and immune cells within the TME can lead to a situation where cancer cells evade recognition by T cells and can influence the response to immune checkpoint blockade (ICB) therapy. Tumor cells and immune cells interact in the cancer microenvironment through inhibitory signaling molecules, and the release of immune suppression by ICBs activates the immune system, leading to an inflammatory response at tumor sites. Tumor gene mutations, such as those in the RAS/Raf/MEK/MAPK pathway, EGFR, and JAK genes, have been linked to ICB response. Additionally, high tumor mutational load and low intratumoral mutational heterogeneity have been correlated with better responses to ICBs. However, while promising, the field is still developing, and further research is needed to fully understand the mechanistic basis for the relationship between immune checkpoint therapy resistance and new genetic mutations [27].

### 5.4. Upregulation of Inhibitory Molecules

Tumor cells might increase the expression of additional inhibitory molecules beyond the immune checkpoints targeted by therapy, further dampening T-cell responses. A resistance mechanism observed in tumor cells involves the elevation of alternative checkpoint inhibitors in response to monoclonal antibody therapy. For instance, in the context of investigating melanoma or prostate cancer, it was discovered that tumors that initially increased their levels of CTLA-4 and were later treated with anti-CTLA-4 antibodies subsequently raised their VISTA expression. This resulted in the activation of a distinct pathway for suppressing T-cell activity. Tumor cells can adapt by upregulating other immune checkpoints, such as VISTA, when one checkpoint is targeted, leading to alternative pathways for suppressing T-cell responses [29].

### 5.5. Upregulation of Cancer Promoting and Immunosuppressive Signaling Pathways in Cancer and Immune Cells Which Involve in ICI Resistance

Resistance to immune checkpoint therapy can arise from the activation of alternative cancer-promoting signaling pathways and the upregulation of additional immunosuppressive signaling pathways [28]. Numerous oncogenic signaling pathways within cancer cells contribute to resistance against immune checkpoint therapy, including receptor tyrosine kinases (RTKs), MAPK, PI3K-AKT-mTOR, JAK-STAT, Hippo, and Wnt pathways [6]. Suppressive functions of immune checkpoints usually depend on ligand-induced signaling. There are several immune-suppressive signaling pathways involved in immune checkpoint therapy resistance, including PD-1 signaling, CTLA-4 signaling, TIM3 signaling, LAG3 signaling, TIGIT signaling [7], IDO (Indoleamine 2,3-dioxygenase) pathway [15], A2AR (Adenosine A2A Receptor) signaling pathway [16,17], and CD73-adenosine pathway [18].

## 6. Role of SHP2 in Checkpoint Therapy Resistance in Immune and Cancer Cells

### 6.1. The Immune Suppressive Effects of SHP2 in the TME

SHP2 can affect the TME by promoting immunosuppressive factors, such as cytokines and chemokines, that create an environment less conducive to effective immune response. SHP2’s role in TME extends to modulating TGF-beta signaling, a crucial pathway in cancer progression. SHP2 can enhance TGF-beta-induced epithelial-mesenchymal transition (EMT) and contribute to immunosuppression within the TME [8]. Indeed, IL-10, an anti-inflammatory cytokine, has an active role in the enhanced antitumor immunity induced by SHP2 or PD-1 targeting in myeloid cells. SHP2 and the PD-1-SHP2 signaling interfere with GM-CSF-mediated phosphorylation of specific proteins, affecting myeloid differentiation. The presence of the PD-1-SHP2 axis in myeloid cells is akin to its established role in B and T lymphocytes. This myeloid-specific PD-1-SHP2 axis could be critical in cancer, where cancer cells can stimulate the production of growth factors and increase PD-1 and PD-L1 expression in myeloid progenitors. Moreover, SHP2 ablation resulted in differentiated neutrophils and tumor-associated macrophages (TAMs) with enhanced effector characteristics, thus promoting antitumor immune responses. Ultimately, SHP2 appears to play a significant role in orchestrating myeloid cell responses within the TME, affecting antitumor immunity and potentially offering avenues for cancer immunotherapy strategies [9].

The immune system fights tumors by producing IFN-γ (a cytokine that stimulates the activation of immune cells) and activating T cells [31]. The inhibitory receptor PD-1 hinders T-cell activation by enlisting the enzyme SHP2 [9]. It is shown that blocking SHP2 has the potential to trigger T-cell help (Th1) immunity, stimulate T-cell activity and eliminate the immunosuppressive effect of cancer [14]. In addition, SHP2 is associated with the skewing of tumor-infiltrating T cells, particularly leukemia Tc1/Th1 cells, towards an inhibitory phenotype, suggesting its involvement in immune regulation within the TME [32].

In natural killer (NK) cells, cell surface inhibitory receptors that activate SHP2 through the ITIM motif were found [33]. Furthermore, SHP2-deficient NK cells produced more IFN-γ in response to tumor target cells [34].

In macrophages, SHP2 regulates multiple signaling pathways, including the CSF-1/CSF-1R axis and CD47/SIRPα, promoting tumor growth and immune evasion in TME. Inhibiting SHP2 in macrophages may promote M1 TAM (antitumor phenotype) polarization and create a microenvironment conducive to antitumor immunity, making it a potential target for tumor immunotherapy [35,36].

SHP2’s potential contribution to shaping the tumor microenvironment (TME) to be less receptive to immune checkpoint therapies. SHP2’s role in fostering immunosuppressive factors could establish an environment that hampers the effectiveness of immune checkpoint inhibitors, contributing to therapy resistance. The promotion of immunosuppressive factors by SHP2 may hinder the desired immune response triggered by immune checkpoint inhibitors, highlighting the intricate interplay between SHP2-mediated TME modulation and immune checkpoint therapy outcomes. Altogether, SHP2 plays a crucial role in maintaining an immunosuppressive microenvironment by suppressing T-cell activation and enhancing the activation of tumor-promoting M2 macrophages.

### 6.2. The Role of SHP2 in Tumor Antigen Presentation in Cancer Cells

SHP2 can impact antigen presentation by affecting MHC expression on tumor cells, potentially limiting their recognition by immune cells. Inhibiting SHP2 in cancer cells enhanced the signaling of IFNγ, leading to a rise in the expression of its downstream targets, such as chemoattractant cytokines and antigen-presenting machinery. Moreover, the genetic removal of PTPN11 within tumor cells led to an increased presence of MHC class I molecules in co-culture scenarios. Conversely, introducing a drug-resistant SHP2 mutant into tumor cells did not elevate MHC class I levels when treated with SHP099, confirming that the heightened IFNγ signaling induced by SHP099 in cancer cells is directly attributable to the targeted inhibition of SHP2. Since deficiencies in presenting antigens are linked to resistance against T-cell-triggered tumor elimination, the enhancement of MHC class I expression on tumor cells through SHP2 inhibition offers a valid rationale for synergizing the SHP2 blockade with immunotherapy for cancer patients [12].

### 6.3. Role of SHP2 in Tumor-Promoting and Immunosuppressive Signaling Pathways in Cancer and Immune Cells

#### 6.3.1. Immune Roles of SHP2 in Immunosuppressive Signaling Pathways

##### SHP2 in PD-1 Signaling, RAS/ERK and PI3K/AKT Signaling

PD-1 is crucial in dampening immune reactions and fostering self-tolerance by regulating T-cell function, triggering apoptosis of T cells specific to antigens, and restraining the apoptosis of regulatory T cells [37]. Within the PD-1 signaling pathway, SHP2 plays a pivotal role by participating in intricate regulatory mechanisms that modulate T-cell receptor (TCR) signaling and downstream events. PD-1 engagement orchestrates the recruitment of SHP2 in proximity to the T-cell receptor, leading to the inhibition of proximal kinase activation and subsequent attenuation of Lck-mediated phosphorylation of ZAP-70. This interaction results in the dampening of key signaling cascades, particularly the PI3K-Akt and Ras-MEK-ERK pathways. In the context of the PI3K-Akt pathway, PD-1 employs SHP2 to block PI3K activation, whereas it directly inhibits Akt activation. Pertaining to the Ras-MEK-ERK pathway, PD-1 interferes with the activation of PLC-γ1 and Ras, thereby impairing MEK-ERK-MAP kinase pathway activation. Moreover, a possible explanation is that SHP2 is sequestered by PD-1, leading to the outcome that SHP2 loses its capability to remove inhibitory phosphate groups from signaling molecules like Lck and subsequent components engaged in the Ras/Erk pathway. SHP2’s presence in the PD-1 pathway serves to curtail T-cell activation through multiple mechanisms. While SHP2’s preference for CD28 over TCR is evident, PD-1-mediated recruitment of SHP2 appears to target both pathways, blocking PI3K activation and suppressing TCR signaling by inhibiting the phosphorylation of TCR and downstream molecules such as ZAP70. Alongside its inhibitory functions in T-cell signaling, SHP2 was also documented to activate TCR signaling by reversing the inhibitory phosphorylation of LCK caused by CSK. Evidence indicates that the seclusion of SHP2 due to phosphorylated PD-1 prevents its ability to enhance LCK activity, thereby contributing to the suppression of T-cell signaling. PD-1 plays a role in regulating T cell–dendritic cell interactions, contributing to the modulation of T-cell activation dynamics. PD-1 engagement with PD-L1 or PD-L2 triggers inhibitory signaling through its cytoplasmic domain, involving ITIM and ITSM motifs that recruit SHP2. The intricate interplay of SHP2 within the PD-1 pathway showcases its multifaceted influence on T-cell activation and immune response regulation, emphasizing its significance as a key mediator of immune checkpoint modulation [7,38,39].

The inhibition of SHP2 within the PD-1 signaling pathway has emerged as a promising strategy in the context of immune checkpoint inhibitor (ICI) resistance therapy. SHP2, a critical regulator of PD-1 signaling, plays a dual role in immune response modulation: it can dampen T-cell activation by interfering with TCR and co-stimulatory signaling while also stimulating TCR signaling through LCK activation. This dual role underscores the complexity of SHP2’s involvement in immune regulation. Leveraging SHP2 inhibition as a therapeutic approach involves addressing its context-dependent functions. In the case of ICI resistance, targeting SHP2 might counteract the dampening effect on T-cell activation and enhance the response to checkpoint inhibitors. However, the precise modulation of SHP2’s inhibitory and stimulatory effects will be crucial to avoid potential adverse effects. As research continues to elucidate the intricate mechanisms of SHP2’s involvement in the PD-1 pathway, novel therapeutic strategies that harness its potential could pave the way for more effective ICI resistance therapies, offering renewed hope for cancer patients facing limited treatment options.

##### SHP2 in CTLA-4 Signaling

SHP2 emerges as a significant mediator in the CTLA-4 signaling pathway, contributing to the intricate regulation of T-cell activation. Upon T-cell activation, the phosphorylated YVKM motif in CTLA-4’s cytoplasmic tail is implicated in recruiting SHP2, which in turn aids in repressing T-cell activation. This recruitment of SHP2 potentially forms a part of CTLA-4’s mechanism for delivering inhibitory signals. The phosphorylated YVKM motif of CTLA-4 recruits SHP2 to inhibit RAS. CTLA-4 also inhibits AKT activity through PP2A. This finding underscores SHP2’s dual role in immune regulation, as observed in other signaling pathways. While the exact extent of SHP2’s involvement in the CTLA-4 pathway is still being elucidated, its potential interaction with CTLA-4 highlights its significance as a common regulatory element across various immune checkpoint pathways. The interplay between CTLA-4, SHP2, and other associated molecules in orchestrating immune responses adds another layer of complexity to the modulation of T-cell activation and the broader context of immunoregulation [7,40].

Targeting SHP2 inhibition within the CTLA-4 signaling pathway holds potential for overcoming immune checkpoint inhibitor (ICI) resistance. By disrupting the interplay between SHP2 and CTLA-4, this approach aims to enhance T-cell activation and counteract inhibitory signaling, offering a strategy to restore ICI responsiveness. However, careful modulation is crucial to balance immune activation while preventing potential adverse effects, highlighting SHP2’s significance in designing effective therapies for ICI resistance.

##### SHP2 in BTLA Signaling

Within the BTLA (B- and T-lymphocyte attenuator) signaling pathway, SHP2’s role is intertwined with the intricate mechanisms governing immune regulation. BTLA, possessing ITIM and ITSM motifs in its cytoplasmic domain, employs these motifs to orchestrate inhibitory signaling. Upon ligand engagement, both tyrosine residues within these motifs become phosphorylated, facilitating the recruitment of SHP1 and SHP2. Notably, BTLA exhibits a preference for recruiting the potent phosphatase SHP1, distinguishing it from PD-1 signaling, which mainly recruits SHP2. This unique recruitment of SHP1 by BTLA contributes to the effective dampening of TCR and CD28 signaling, ultimately regulating T-cell responses. Furthermore, BTLA’s role extends to T follicular helper (Tfh) cells, where its engagement with HVEM on B cells leads to the recruitment of SHP1, culminating in the inhibition of TCR signaling and modulation of B cell proliferation. This intricate interplay underscores SHP2’s role in fine-tuning immune responses through the BTLA pathway, shedding light on potential avenues for immune regulation modulation [7].

By disrupting SHP2-mediated inhibitory signaling downstream of BTLA engagement, this strategy aims to enhance T-cell activation and counteract resistance mechanisms, potentially restoring ICI effectiveness. Careful modulation of SHP2 activity holds the potential for overcoming ICI resistance, offering a novel avenue for improving therapeutic outcomes.

##### SHP2 in TCR Signaling and RAS/MAPK Signaling Pathways

T-cell receptor (TCR) signaling leads to T-cell activation, proliferation, and the immune response. SHP2, a critical player in TCR signaling, has a complex role in the pathway, exhibiting both positive and negative regulatory effects depending on the context and signaling molecules involved. Acting as a dual-specificity phosphatase, SHP2 modulates the phosphorylation status and activity of key molecules, such as Zap70 and Lck, integral to TCR-mediated signaling. It can positively regulate TCR signaling by activating downstream molecules like Ras-ERK, which is crucial for T-cell proliferation and differentiation. However, SHP2 also negatively regulates the pathway either due to direct dephosphorylating signaling molecules such as ZAP-70, a key kinase in TCR signaling, or due to the inhibition of TCR signaling via PD-1. This multifaceted involvement of SHP2 shapes T-cell activation and immune responses. In addition, SHP2 contributes to immune regulation via inhibitory receptors. It interacts with inhibitory receptors like PD-1 and BTLA through ITIMs and ITSM, adding complexity as its inhibitory role intersects with positive contributions to TCR signaling. SHP2 deficiency in T cells can lead to impaired TCR signaling and defective immune responses. The interplay of SHP2 with PD-1, including feedback mechanisms, and its broader impact on TCR signaling and immune regulation continue to be active areas of research, influencing our understanding of T-cell responses, especially in the context of cancer immunotherapy [3,39,41].

##### SHP2 in BCR Signaling

When PD-1 is co-engaged with BCR, it results in the phosphorylation of both tyrosine residues in PD-1. SHP2 is then recruited to the C-terminal phosphotyrosine of PD-1 and is subsequently phosphorylated. The phosphorylated SHP2 then acts to dephosphorylate proximal signal transducers of BCR, including molecules like Syk and Igα/β. This deactivation of downstream molecules, such as PI3K, PLCγ2, and ERK, leads to inhibiting acute-phase reactions like Ca^2+^ mobilization and long-term effects, such as growth retardation. This suggests that SHP2 plays a direct role in regulating BCR signaling through its involvement in the PD-1 pathway, contributing to the modulation of B-cell activation and responses [42].

##### SHP2 in NK Cell Signaling

It is shown that SHP2 is associated with NK (natural killer) cell signaling, and SHP2 expression negatively regulates NK cell function. One study describes a newly identified role for SHP2 in dampening NK cell activation, especially in response to tumor target cells. Using various experimental approaches involving knockdown and overexpression of SHP2 in NK-like cell lines, the study demonstrates that SHP2 suppresses cytolytic activity and cytokine production in a concentration-dependent manner. This inhibitory effect impacts MTOC (microtubule organizing center) polarization and granzyme B release during NK cell responses to target cells. Interestingly, SHP2 overexpression reduced NK-target cell conjugation, while SHP2 silencing did not influence this process. The findings highlight SHP2’s role as a general inhibitor of NK cell responsiveness, shedding light on its involvement in regulating NK cell functions and potential implications for therapeutic interventions targeting SHP2 to enhance NK cell activity in cancer or viral infections [43].

##### TLR Signaling

SHP2 is directly associated with TLR (Toll-like receptor) signaling. Belonging to the SH2-domain-containing family of non-membrane protein tyrosine phosphatases, SHP2 shares sequence identity with SHP-1 but has ubiquitous expression and regulates diverse signaling pathways. Notably, SHP2 has been found to inhibit IFN production in response to TLR3 and TLR4 ligands, as demonstrated by An et al. SHP2 deficiency led to enhanced IFN-β expression upon TLR activation, with direct interaction between SHP2 and the kinase domain of TBK1 inhibiting IRF3 activation and IFN production. Xu et al. also confirmed that SHP2^−/−^ macrophages displayed increased IFN-β secretion upon TLR activation. Furthermore, SHP2 was shown to modulate MAPK pathways, with its deletion attenuating JNK and p38 MAPK activation in response to TLR stimulation in various cell types. The exact mechanisms underlying these interactions warrant further investigation, possibly involving TRAF6 ubiquitination and TAK1 activity modulation. Additionally, evidence suggests that SHP2 might be implicated early in TLR engagement, as active SHP2 mutant expression in endothelial cells correlated with reduced LPS-induced barrier disruption and increased FAK phosphorylation [44].

##### SHP2 in JAK/STAT Pathway in Cytokine Receptor Signaling

The JAK/STAT signaling pathway, which plays a crucial role in immune cells, regulates gene expression and mediates responses to various cytokines and growth factors, thus significantly influencing immune function and inflammation. This pathway also orchestrates the immune system, particularly in the polarization of T helper cells, and is tightly controlled by proteins like Suppressors of Cytokine Signaling (SOCS), Protein Tyrosine Phosphatases (PTPs), and Protein Inhibitors of Activated STATs (PIAS). Dysregulation of this pathway can lead to various immune disorders. Among the regulators, SOCS proteins, SHP1 and SHP2, negatively modulate JAK-STAT activity through mechanisms such as direct dephosphorylation of JAKs or competitive binding to phosphorylated receptors. These regulators ensure precise initiation, duration, and termination of the signaling cascade, preventing uncontrolled immune responses. The balance of these regulators finely tunes the immune response, and understanding their roles opens avenues for potential therapeutic interventions in immune-related diseases. Additionally, other post-translational modifications like SUMOylation can further influence the activity of STAT proteins. Understanding these regulatory mechanisms offers the potential for targeted therapeutic interventions to treat immune-related diseases and disorders [45].

#### 6.3.2. Non-Immune Roles of SHP2 in Oncogenic Pathways (Tumor-Promoting) in Cancer Cells

SHP2 participates in several oncogenic signaling cascades, including RAS/MAPK, PI3K/AKT, and JAK/STAT pathways (Figure 2) [5,14,46]. In addition, MAPK, PI3K-AKT-mTOR, and JAK-STAT pathways in cancer cells play a crucial role in ICI resistance [6]

Targeting SHP2 inhibition holds promise for combating immune checkpoint inhibitor (ICI) resistance by modulating critical signaling pathways. SHP2 inhibition could disrupt the MAPK, PI3K-AKT-mTOR, and JAK-STAT pathways, which are crucial for cancer cell survival and immune evasion. SHP2 inhibition has been suggested to attenuate MAPK pathway activation, impacting cell proliferation and survival. Moreover, SHP2 inhibition may hinder PI3K-AKT-mTOR signaling, curbing cancer cell growth and proliferation. Additionally, SHP2 inhibition has the potential to impact JAK-STAT signaling, which is critical for immune evasion mechanisms and tumor-mediated immunosuppression. These strategies targeting SHP2’s role in these signaling pathways could provide innovative approaches to enhance the efficacy of immune checkpoint blockade against resistant cancer cells.

##### RAS/ERK/SOS1

In addition to acting as an adaptor protein (activates RAS-RAF-ERK signaling by bridging upstream and downstream signals), SHP2 can also dephosphorylate downstream signaling molecules, activating downstream effectors in biological processes, including cell proliferation, migration, metabolism, and differentiation [5]. Following growth factor stimulation, SHP2 is recruited by RTKs (SH2 domains link to the RTKs) and can function as an adaptor protein by attaching to phosphotyrosine-binding substrates like GAB1/2, GRB2, IRS1, FRS2, and Shc resulting in ERK activation [14].

Grb2/SOS complex could be recruited to the tyrosine phosphorylation sites on SHP2 (Tyr542 and Tyr580), and RAS is activated in an SOS-dependent manner, leading to downstream stimulation of Raf1, MEK1/2 (intermediate kinases), and ERK1/2 (Raf1 downstream target). As a result, stimulated ERK1/2 induces the activity of several effectors, including transcription factors, which directly contribute to the proliferation and differentiation of cells [47,48]. Through the activation of Src family kinases, SHP2 could indirectly activate RAS. Recruitment of SHP2 to Gab1 leads to dephosphorylation of paxillin and Cbp/PAG and the disconnection of Csk (a negative regulator of paxillin and src) and Src, and finally, results in src activation [49].

##### PI3K/AKT

The crucial signaling system known as PI3K/AKT controls biological and physiological responses such as cell growth, survival, homeostasis, and metabolism [48,50]. After activating EGFR, it recruits Gab1 through SHP2. Then, phosphorylated tyrosine Gab1 attracts PI3K and encourages the synthesis of PIP3 locally. Next, PIP3 can translocate Gab1 to the plasma membrane close to EGFR, which causes Gab1 to become phosphorylated and activates PI3K. Activated PI3K mediates the conversion of PI (4,5)P2 to PI (3,4,5)P3, which then employs AKT and PDK1, leading to cell survival and growth [51]. Moreover, SHP2 can down-regulate PI3K activation by dephosphorylating Gab1 [52] and activate IRS1 through its phosphorylation [14]. Thus, it seems that SHP2 has opposing functions in regulating the PI3K-AKT-mTOR pathway, depending on the stimuli.

##### JAK/STAT

STAT proteins, a family of transcription factors, are crucial for several biological processes, including cell growth, cell survival, differentiation, anti-apoptosis and cell motility [53]. Based on substrate specificity, SHP2 has positive and negative functions in the JAK/STAT pathway. JAK2 activity and STAT5 phosphorylation are both reduced in SHP2-inactivated cells. SHP2 can dephosphorylate JAK and block the interaction of JAK with SOCS, reactivating the STAT signaling pathway [14,54]. As a negative effect, the overexpression of SHP2 enhances STAT5’s dephosphorylation level in response to IL-3 stimulation, inhibiting STAT5 activity [55,56]. The gp130 receptor typically transmits the JAK-STAT signaling that IL-6 generates, and mice lacking SHP2 showed prolonged activation of STAT3 by the gp130 receptor, suggesting a negative function of SHP2 [57]. The IL-6/IL-6R/gp130 complex recruits SHP2 and binds to the cytoplasmic tyrosine tail of gp130 to modulate downstream pathways, including dephosphorylation of STAT3 to negatively control the JAK-STAT3 signaling pathway and activating STAT3. This leads to the synthesis of PRL-3, which ultimately suppresses the activation of SHP2 [58].

## 7. SHP2 Inhibitors and Their Clinical Development in Oncology

Gain of function mutations of SHP2 plays a key role in the development and advancement of tumors and cancer by affecting cells directly [11]. On the other hand, SHP2 plays a crucial role in numerous tumor-promotive and immune-suppressive signaling pathways within TME, cancer, and immune cells. Therefore, the inhibition of SHP2 can serve as a therapeutic approach.

### 7.1. SHP2 Inhibitors

SHP2 inhibitors generally include allosteric and orthosteric inhibitors [59]. Orthosteric products bind at the active site, competing with the natural substrate or ligand and block the site from activation [14,59]. Natural products like Tautomycetin (TTN) have shown promise as immunosuppressive and antitumor agents by selectively inhibiting SHP2 and reducing SHP2-dependent signaling. Suppressing SHP2 phosphatase with TTN may reduce SHP2-dependent signaling by reducing ERK1/2 activation [59]. Other orthosteric inhibitors, such as cefsulodins, phenylhydrazonopyrazolone sulfonate derivatives, quinoline hydrazine derivatives, salicylic acid derivatives, diterpenoid quinone derivatives, and oxindole derivatives, have been studied for their potential inhibitory effects on SHP2 [59,60,61,62,63,64,65].

Conversely, allosteric inhibitors target sites other than the active site and change the binding site conformation. They have demonstrated advantages in selectivity and bioavailability compared to orthosteric inhibitors [14]. Four allosteric sites exist, including the “tunnel” at the C-SH2/PTP domain interface. SHP099, SHP389, SHP394, SHP836, BBP-398, and TNO155 stabilize SHP2’s closed auto-inhibited state by attaching to the tunnel-like pocket; the “latch” at the N-SH2/PTP domain interface, such as SHP244 and its derivatives, SHP844 and SHP504; the “groove” at the N-SH2/PTP domain interface, which was on the other side of the tunnel and nonconserved cysteine residue 333 (Cys333) site, which is located in the PTP Domain [66,67,68]. It should be noted that no allosteric inhibitors of SHP2 targeting the Groove-like site have been found [14]. Overall, both orthosteric and allosteric inhibitors offer promising avenues for the development of targeted therapies against diseases involving SHP2-dependent signaling pathways. Some other SHP2′s allosteric inhibitors are listed below [Table 1].

### 7.2. SHP2 Inhibitors in the Clinic

While SHP2 inhibition as monotherapy has shown single-agent activity in solid tumors, a combinatorial approach synergizing known relevant pathways could overcome or delay adaptive resistance in tumors with certain oncogenic drivers. There are 11 small-molecule SHP2 inhibitors in mono- or combination clinical studies being evaluated for the treatment of cancer patients [Table 2].

#### 7.2.1. TNO155

The SHP2 allosteric inhibitor TNO155, created by Novartis Institutes, is highly potent and selective, like other allosteric tunnel inhibitors, exhibited remarkable selectivity across various phosphatases and kinases [77]. TNO155 has been found to be an effective agent for blocking both tumor-promoting and immune-suppressive receptor tyrosine kinase (RTK) signaling in RTK- and MAPK-driven cancers and their TME. This suggests that TNO155 has a dual role in inhibiting SHP2 in both cancer cells and immune cells (inhibits immunosuppressive macrophages), which could potentially enhance the effectiveness of immune checkpoint therapy. Together with EGFR, BRAF, KRAS^G12C^, CDK4/6 inhibitors, or anti-PD-1 antibodies, TNO155 has shown effectiveness and synergistic pathways in pre-clinical studies [78]. Novartis Pharmaceuticals conducted a first-in-human (FIH) trial of TNO155 in adult patients with RTK-dependent solid tumors in May 2017 to evaluate its safety and tolerability both as a monotherapy and in combination with the third-generation EGFR inhibitor EGF816 (Nazartinib^®^) [79]. In a phase 1b clinical study, TNO155 is also being tested with spartalizumab (an anti-PD1 antibody) or ribociclib (a CDK4/6 inhibitor) in oncology [80]. To identify the maximum tolerated dosage and recommended dose in adult patients with advanced or metastatic BRAF V600 CRC, a phase 1 study of TNO155 in combination with Dabrafenib (Tafinlar^®^, BRAF inhibitor) and LTT462 (ERK inhibitor) was launched in 2020 [81]. Patients with advanced solid tumors with KRAS^G12C^ mutations are also being recruited for a phase 1/2 trial of TNO155 in combination with MRTX849, a selective inhibitor of KRAS^G12C^ [82]. Further, a KRAS^G12C^ inhibitor, JDQ443, is now being studied in advanced solid tumor patients in a phase 1/2 study of TNO155 and Spartalizumab and is now showing promising signs of efficacy in refractory patients with KRAS^G12^ mutated tumors, when given in combination with a KRAS^G12^ inhibitor [83].

#### 7.2.2. RMC-4630

The SHP2 allosteric inhibitor RMC-4630 is a potent and selective compound To treat solid tumors with genetically specified mutations in SHP2-dependent signaling pathways, such as KRAS^G12C^, NF1LOF, and BRAF class 3. In early pre-clinical tests, RMC-4630 was shown to slow the growth of tumors and kill tumor cells by reducing the signaling from the Ras-Raf-MEK-ERK cascade in human cancers with specific mutations in this pathway [84,85].

Amgen has started a phase 1b clinical study in patients with advanced solid tumors and the KRAS^G12C^ mutation to test the safety and effectiveness of AMG510 (a covalent mutant selective KRAS^G12C^ inhibitor) in combination with RMC-4630. In addition, there are synergistic antitumor effects when RMC-4630 and cobimetinib (a MEK inhibitor) are used together in treatment. The Phase 1b/2 study of RMC-4630 in combination with cobimetinib in adult patients with relapsed or refractory solid tumors harboring specific genomic mutations started in July 2019, and finally, phase 2 was completed in February 2022 was [86].

RMC-4630 and LY3214996 (an ERK inhibitor) were combined to inhibit SHP2 upstream of KRAS. The safety of the RMC-4630 and LY3214996 combination therapy in treating CRC, NSCLC, and pancreatic ductal adenocarcinoma will be examined in the phase I study. The study started in March 2022, and its estimated completion date is July 2024 [87].

The safety and effectiveness of RMC-4630 in combination with sotorasib (a KRAS^G12C^ inhibitor) in patients with KRAS^G12C^ mutant NSCLC are being assessed in a phase 2 open-label study. The study started in December 2021, and its estimated completion date is January 2024 [88,89].

Adult participants with locally advanced or metastatic EGFR mutation-positive NSCLC participated in an open-label, phase 1b/2 study to determine the safety, maximum tolerated dose (MTD) of RMC-4630 in combination with Osimertinib (a tyrosine kinase inhibitor). The study started in July 2019 and finally started in February 2022 and was completed [86].

RMC-4630 and pembrolizumab are used in an open-label multicenter phase 1 research to assess the safety in patients with solid tumors in part 1. The anticancer activity and safety of RMC-4630 in combination with pembrolizumab (a PD-1 inhibitor) in people with metastatic 1L lung cancer will be evaluated in Part 2 of the expansion cohort. Participants with lung cancer and a KRAS^G12C^ mutation will be reviewed in Part 3 for the safety, MTD, and antitumor efficacy of RMC-4630 combined with adagrasib (A KRAS^G12C^ inhibitor). Part 4 will determine how the food and formulations affect RMC-4630’s PKs when given pembrolizumab. The study began in June 2020, is still ongoing, and is anticipated to be finished in February 2022 [90,91].

#### 7.2.3. Sodium Stibogluconate

Sodium stibogluconate (SSG, Lenocta, and VQD-001) is being studied in humans as the SHP2 inhibitor to target the catalytic region of the protein [92]. According to pharmacological research, SSG may enhance the inverse effects of GM-CSF and IFN- on TF-1 cell development, demonstrating its extensive activities in signaling different cytokine responses in hemopoietic cell lines [92]. The MD Anderson Cancer Center initiated a phase I, open-label research in September 2006 to assess SSG’s safety PKs in combination with IFN-2b for patients with advanced malignancies, but the study was stopped four years later [93].

#### 7.2.4. JAB-3312 and JAB-3068

The solid metastatic tumors of the head and neck, esophagus, and other areas can all be treated with JAB-3068 [94,95]. SHP2, as a downstream effector of the PD-1 signaling pathway, can alleviate cancer-induced immunosuppression in the TME. JAB-3068 has the potential to reduce tumor burden by enhancing CD8+ antitumor immunity through cytotoxic T-cells, indicating possible therapeutic promise for NSCLC, CRC, and esophageal squamous cell carcinoma (ESCC) when used in combination with an anti-PD1 antibody. The KRAS-MAPK signaling pathway inhibitor JAB-3312 has shown promise in treating a wide variety of solid tumors, including NSCLC, CRC, and pancreatic cancer. Like JAB-3068, JAB-3312 may improve the effectiveness of current tumor immunotherapies by reducing the immunosuppressive milieu around tumors [5]. In 2019, the FDA and Chinese regulators agreed to allow JAB-3312 to enter clinical studies in humans with advanced solid tumors [96,97]. The study completion date will be March 2024 [97].

#### 7.2.5. ERAS-601

This drug is a potent SHP2 inhibitor. In 2020, Erasca launched a phase 1/1b clinical trial of ERAS-601 as a monotherapy and in combination with a MEK inhibitor in patients with advanced or metastatic solid cancers carrying specific mutations [98].

#### 7.2.6. BBP-398

BBP398, an allosteric SHP2 inhibitor, showed no substantial inhibition against a panel of 22 phosphatases, including SHP1, which revealed its excellent selectivity [59]. Both in vitro and in vivo studies on EGFR mutant osimertinib-resistant NSCLC models demonstrated that BBP-398, alone or in combination with osimertinib, an EGFR inhibitor, exhibited a potent tumor-suppressing activity. Evidence suggests that BBP-398 may re-establish osimertinib sensitivity in NSCLC models resistant to the drug [99]. The phase 1 study to test BBP-398 in solid cancers with gene alterations in the MAPK signaling system, including Ras and RTKs, was launched in 2020 [100]. The study focuses on advanced KRAS^G12C^ mutant NSCLC, advanced KRAS^G12C^ mutant non-NSCLC, solid tumors with other MAPK pathway mutations, and advanced EGFR-mutant NSCLC [5].

#### 7.2.7. RLY-1971

In order to determine the safety, PKs, and preliminary antitumor efficacy of RLY-1971 (a potent SHP2 inhibitor), an open-label trial has been underway since January 2020 in patients with advanced or metastatic solid tumors. Relay Therapeutics and Genentech have planned various combination studies of RLY-1971 with inhibitors of KRAS^G12C^ and GDC-6036 [5].

#### 7.2.8. HBI-2376

HBI-2376 is a selective SHP2 inhibitor that may be used for treating solid tumors with KRAS or EGFR mutations, such as NSCLC and CRC. Further, HBI-2376 may inhibit tumor development by stimulating immune infiltrating cells in the TME. HBI-2376 inhibited tumor development more effectively than TNO-155 and RMC-4550, both as a single drug and in combination, as shown by xenograft results. With HBI-2376’s favorable safety profile, the FDA granted IND approval. Future studies using this drug in patients with NSCLC or CRC could be promising [101].

#### 7.2.9. BPI-442096

The small chemical BPI-442096 is a potent and selective SHP2 inhibitor. Multiple KRAS mutant cancer cell lines, such as those from NSCLC, PDAC, CRPC, etc., were significantly inhibited from proliferation with this compound. BPI-442096 suppressed SHP2 phosphatase and subsequent ERK phosphorylation in cancer cells and NFAT reporter gene expression after PD-1/PD-L1 signaling in immune cells. BPI-442096 significantly suppressed tumor development in xenograft mice models with KRAS^G12C^, KRAS^G12D^, and KRAS^G12V^ mutations. In addition, in the MC38 syngeneic model, BPI-442096, either alone or combined with anti-PD1/PD-L1 medicines, induced antitumor immunity. In addition, BPI-442096, in combination with KRAS^G12C^ inhibitor, may overcome innate and acquired resistance to KRAS^G12C^ inhibition. BPI-442096 has many potential combinations to overcome drug resistance, and it has potent antitumor activity in several KRAS mutant models and increased anticancer immunity in syngeneic mice models [102].

#### 7.2.10. SH3809

SHP3809 has been shown to inhibit SHP2 and tumor development in NCI-H358 and KYSE520 cancer cells both in vitro and in vivo, opening up a potential treatment route for patients with solid tumors. Phase I, open-label research of the SH3809 for treating advanced solid tumors, began in April 2021 [103].

## 8. Overcoming Drug Resistance in Solid Tumors: SHP2 Inhibitors in Combination with Immune Checkpoint Inhibitors

Recent advancements in cancer research have focused on addressing drug resistance. To combat this challenge, researchers have explored innovative treatment approaches, and one promising strategy involves using SHP2 inhibitors in combination with immune checkpoint inhibitors such as PD-1, RTK, MEK, ERK, and ALK inhibitors. Activation of alternative receptor tyrosine kinases (RTKs) and downstream signaling cascades, such as the PI3K/AKT pathway, has been implicated in immune checkpoint therapy resistance [104,105,106]. SHP2, as a downstream effector of multiple RTKs, plays a crucial role in these signaling networks. SHP2 plays a significant role in drug resistance mechanisms, making it an attractive therapeutic target [14]. The deregulation of SHP2 is a frequent resistance mechanism in targeted treatments since it may operate as an oncogenic factor or tumor suppressor in various malignancies. Moreover, SHP2 regulates the suppression of immunological checkpoints, inhibiting patients’ antitumor immune responses, and making it an attractive therapeutic target [104]. In general, many advanced tumors have more than one altered pathway, and combination strategies matched to the alterations present are associated with better outcomes [107,108,109]. With SHP2, combination therapy has shown greater success than monotherapy in preclinical studies, overcoming drug resistance and issues arising from monotherapy [110]. SHP2 inhibitors have demonstrated potential in addressing resistance to kinase inhibitors and PD-1 blockade [14]. Resistance to TKIs can arise from various factors, including genetic alterations, gene amplification, and protein expression changes [111]. TKIs can impair the function of ATP-binding cassette (ABC) transporters, making them useful in combination treatments. MDR-related ABC transporters regulate intracellular concentrations of small-molecule inhibitors. Combining SHP2 inhibitors with other protein inhibitors can overcome drug resistance [14]. Inhibiting SHP2 provides a novel means of overcoming resistance mechanisms in cancer therapy, opening new possibilities for more effective and personalized treatments [78].

SHP2 inhibitors can be logically combined with immunotherapies like checkpoint inhibitors due to their ability to affect anti-tumor activity through both tumor intrinsic and immune-mediated mechanisms [78]. By combining SHP2 inhibitors with immune checkpoint inhibitors, researchers hope to achieve synergistic effects, overcome drug resistance, and improve treatment outcomes for patients with thoracic malignancies. Promising clinical combinations include SHP2 inhibitors with immune checkpoint inhibitors such as PD-1, RTK, MEK, ERK, and ALK inhibitors, as discussed below.

### 8.1. PD-1/PD-L1

PD-1 is found in high tumor-infiltrating lymphocyte (TIL) concentrations and suppresses T-cell activation [112]. High PD-L1 or PD-1 expression in a tumor is linked to tumor immune escape and poor overall prognosis [113,114,115]. The suppression of immune checkpoints, especially PD1 blockade, has revolutionized clinical oncology in the last decade [116,117]. Improving T-cell responsiveness and mediating anticancer activity in preclinical models may be accomplished by blocking the interaction between PD-1 and PD-L1 [118]. The anti-PD-1/PDL-1 therapy has been quite successful, although many patients with solid tumors still showed primary and acquired drug resistance. Tumors may form a TME under PD-1/PDL-1 therapy to inhibit the anti-tumor action of T lymphocytes. Because of its capacity to bind to several immunosuppressive receptors, SHP2 has a powerful tumor-killing impact when its activity is blocked. Researchers found that when used together, SHP2 inhibitor and anti-PD-1 antibody were more effective than either treatment alone in halting tumor development [24]. Hence, future tumor immunotherapy may benefit from developing a particular SHP2 inhibitor in combination with a PD-1 antagonist [31]. For example, the variety of RMC-4630 with Pembrolizumab, BBP-398 with Nivolumab, and TNO-155 with Spartalizumab and Tislelizumab have entered clinical trials [90]. In addition, SHP099 boosts T-cell activity in the mouse model, and the combination of SHP099 and anti-PD-1 could kill tumor cells more efficiently, demonstrating that SHP2 is a potentially viable therapeutic method for cancer therapy [24]. As a result, PD-1 or SHP2 inhibition can restore significant Th1 immunity and T-cell activation, reversing immunosuppression in the TME [31].

### 8.2. RTK Inhibitors

Research shows that SHP2 is highly up-regulated when RTK is activated to gain adaptive resistance [119]. The upstream signal of RTK and SHP2 in cancers with KRAS mutations presents a novel target for RTK and SHP2 inhibitors. By blocking SHP2, the selective allosteric SHP099 has shown that there is hope for treating cancers that depend on RTKs. Inhibiting SHP2 and RTK effectively treats tumors with KRAS mutations like KRAS^G13D^ and KRAS^Q61H^, which rely on upstream growth factor signaling [120]. Several genetic and pharmacological evidence-based studies demonstrate that SHP2 is essential to RTK signals, such as FGFR, VEGFR, PDGFR, and EGFR signals, ultimately activating the whole ERK pathway [49,121]. By reactivating the RAS-MEK-ERK pathway, hepatoma cells develop adaptive drug resistance when treated with Sorafenib (a multi-kinase inhibitor including RTK). The adaptive resistance to sorafenib may be overcome by combining it with SHP099, which inhibits the reactivation of the MEK/ERK signaling pathway. As a potential novel therapy method for hepatocellular carcinoma (HCC), combining SHP099 with sorafenib may considerably increase the survival rate and tumor growth inhibition [119].

### 8.3. MEK Inhibitors

The overactivation of MEK or an inability of the inhibitor to bind to MEK due to mutations in MEK leads to drug resistance. Most malignancies that develop resistance to MEK inhibitors do so because several RTKs upstream of the MAPK pathway are reactivated, triggering a signal cascade that ultimately leads to uncontrolled cell proliferation [122]. As a result, the therapeutic use of MEK inhibitors was limited by the development of adaptive drug resistance [123]. It has recently come to light that SHP2 inhibitors may counteract the adaptive drug resistance seen with MEK inhibitors. So, a novel approach to treating RAS-driven cancer may be to combine MEK and SHP2 inhibitors [124].

Growth factor-limiting conditions demonstrated that SHP2 inhibition or knocking out SHP2 suppressed the stemness of KRAS-mutant NSCLC cells; this effect was amplified by MEK inhibition [125]. In another study, SHP099 was used to treat patient-derived xenograft (PDX) models. SHP2 inhibition considerably inhibited subcutaneously implanted PDX2 KRAS-mutant NSCLC development, and a decrease in pERK levels in tumors accompanied this. These findings show that SHP2 and MEK inhibitors have a synergistic anti-proliferative impact. KRAS mutant cancers, particularly NSCLC, respond well to dual SHP2 and MEK inhibition [126]. Gastric cancer, triple-negative breast cancer (TNBC), and high-grade serous ovarian cancer are particularly resistant to standard treatments; however, the addition of SHP2 inhibitors allows the combination of MEK inhibitors to overcome their adaptive drug resistance [123,126]. In a xenograft model using KRAS^G12C^ NCI-H358 cells, combining the SHP2 inhibitor RMC-4630 with the MEK inhibitor cobimetinib inhibited tumor development with synergistic effects [110]. Pancreatic and colon cancer cell lines are more sensitive to the MEK inhibitor selumetinib (AZD6244) when administered in combination treatment with SHP2 inhibitors [126].

### 8.4. ERK Inhibitors

Specific ERK-dependent cancers may develop adaptive resistance to RAF and MEK inhibitors, but this resistance can be addressed by simultaneously regulating ERK signaling and SHP2 activity. LTT462 is an ERK inhibitor that has antitumor effects. LTT462 suppresses ERK by binding to the protein and blocking its ability to activate signal transduction pathways. As a result, the proliferation and survival of ERK-dependent tumor cells are impacted. LTT462 was recently used in Novartis trials in Q2 2021 and with TNO 155 in CRC patients [127]. Dabrafenib, Trametinib, and SHP099 were used to determine the effectiveness of simultaneous suppression of ERK signal and SHP2 in vivo. This combination dramatically lowered p(Y542) SHP2 and ERK signals in mice harboring RKO xenografts but did not impact body weight. Compared to the combination of dabrafenib and trametinib, this compound seems superior in inhibiting ERK signaling. Tumor growth and ERK signal transduction in xenografts were not significantly affected by dabrafenib, trametinib, or SHP099 treatments alone. Further evidence suggests that blocking the ERK signal and silencing SHP2 may be helpful ways to treat BRAF (V600E) CRCs [14].

### 8.5. ALK Inhibitors

Anaplastic lymphoma kinase (ALK) normally stimulates cell proliferation following ligand binding. ALK inhibitors initially restrain most NSCLCs with an ALK rearrangement; however, SHP2 offers a survival input parallel to several tyrosine kinases that enhance resistance to ALK inhibitors. A recent study indicated that SHP099 alone has little impact on the growth of various tumor cells. When combined with the ALK tyrosine kinase inhibitor ceritinib, the reactivation of RAS and ERK1/2 is blocked, inhibiting drug-resistant patient-derived cell formation. Evidence like this suggests that targeting ALK and SHP2 together might be a practical approach for treating cancer that has developed resistance to standard treatments. Furthermore, short-term and long-term usage of SHP099 alone does not suppress the activity of RAS in any patient-derived tumor cells, but short-term combination administration of ceritinib and SHP099 may diminish the amount of GTP-RAS in all models [128]. MGH045-2A xenografts were entirely resistant to ceritinib therapy, whereas MGH049-1A and MGH073-2B xenografts responded mildly and transiently. The combination of SHP099 and ceritinib, on the other hand, led to a profound regression of MGH049-1A and MGH073-2B xenografts and moderate inhibition of the development of MGH045-2A tumors, which was in keeping with the significantly lower DUSP6 mRNA level, when treatment with SHP099 and ceritinib was discontinued in the MGH073-2B and MGH049-1A animals, tumor cell growth resumed, and resistance to the medicines was maintained. As a result, blocking ALK and SHP2 activity may be a practical therapeutic approach for patients with NSCLC who have developed resistance to treatment [128].

### 8.6. CDK4/6 Inhibitors

CDKs are kinases that control cell cycle and transcription. Cell cycle CDKs move cells through different phases of division. CDK4/6 is crucial for the G1- to S-phase transition and abnormalities in CDK4/6 and the NF1 (a negative regulator of RAS activity) gene may contribute to neuroblastoma and malignant peripheral nerve sheath tumors. SHP2 knockdown or SHP2i therapy reduced CDK4/6i-induced ERK signaling activation and cyclin D1 expression and increased sensitivity to CDK4/6i. Results from both in vitro cell growth and in vivo PDX showed benefit when the two were combined. Furthermore, other studies show tumors with MEK and cyclin alterations respond to the combination of MEK and cyclin inhibitors but not generally to inhibitors of one of the pathways [129,130].

Preliminary evidence indicates that inhibiting SHP2 and CDK4/6 simultaneously is very effective and leads to a long-lasting response in NF1-associated MPNST pre-clinical models. Tumors treated with either SHP2i alone or SHP2i in combination with CDK4/6i showed a reduction in p-ERK levels in pharmacodynamic trials. The treatment of MPNST with this combo approach has the potential to be innovative [131].

### 8.7. BRAF Inhibitors

The oncogenic BRAF kinase dysregulates the ERK signaling pathway in various human cancers. In addition to their molecularly focused action, BRAF inhibitors can have immunomodulatory effects [132,133]. BRAF inhibitors improve the TME and anti-tumor immune response in BRAF-mutant melanoma because the MAPK pathway activates the T-cell receptor [133]. The critical need for improved BRAF inhibitors is highlighted by the observation that RAF inhibitors, in conjunction with BRAF^WT^, show paradoxical activation in normal tissue due to RAF transactivation, acquired drug resistance, and reduced clinical efficacy in non-V600 BRAF-dependent malignancies [132]. In order to better understand how BRAF inhibitors work, several clinical studies targeting different types of malignancies with BRAF mutations are now being conducted [134]. Synergistic effects were shown when SHP2 inhibitors were combined with vemurafenib, and the combination overcame neuroblastoma’s resistance to SHP2 inhibition in vitro and in vivo [72].

## 9. Future Perspectives

Recent research indicates that powerful SHP2 degraders might be able to bypass adaptive resistance by directly breaking down SHP2 [61]. This breakthrough was made possible with the innovative PROTAC technology [5]. PROTACs could potentially outperform traditional SMIs and could be used in conjunction with SMIs to provide synergistic inhibition of SHP2. This is due to PROTAC’s use of differential mechanisms, its ability to target mutant and unforgettable proteins, and better target selectivity [135,136]. However, while PROTACs have numerous advantages over other inhibitors, there are several limitations due to concerns about toxicity.

The emergence of immune checkpoint inhibitors (ICIs) has revolutionized cancer treatment, providing durable responses in a subset of patients across various cancers [10]. However, a significant challenge in the clinical application of ICIs is the development of resistance, leading to treatment failure and disease progression [137]. One potential mechanism by which inhibiting SHP2 can enhance the efficacy of immune checkpoint inhibitors is through modulation of the TME [11]. SHP2 inhibition effectively controls tumor growth by bolstering immune surveillance and facilitating the elimination of cancer cells. This is accomplished through multiple mechanisms, including the amplification of IFNγ signaling, heightened secretion of chemoattractant cytokines to attract anti-tumor lymphocytes, enhanced antigen presentation by cancer cells, positive regulation of CD8 T-cell proliferation and function, and the inhibition of the suppressive effects of immunosuppressive myeloid cells on anti-tumor T cells. This rebalancing of the immune cell composition within the TME can create a more favorable environment for the anti-tumor immune response [12].

Additionally, SHP2 regulates key signaling pathways involved in APC activation and antigen presentation, such as the MAPK and JAK-STAT pathways [14,138]. The phosphorylation and activation of SHP2 in APCs contribute to innate immunosuppression, resulting in a shortage of neoantigens and restricted infiltration of immune cells [13]. The inhibition of SHP2 can enhance antigen presentation by APCs, leading to increased recognition of tumor antigens by T cells and subsequent immune-mediated tumor cell killing [12,13]. Inhibiting SHP2 in T cells can initiate immune responses against tumors. The treatment with SHP099, a SHP2 inhibitor, boosts type I interferon signaling in CD8 T cells and activates NK cells. Additionally, NK cells can be activated through type I interferon-induced IL-15, offering an alternative pathway to target tumors that have low neoantigen or class I MHC expression, evading CD8 T cells. These findings provide valuable insights for the development of enhanced lymphocyte cell therapy that targets SHP2-associated type I interferon signaling. Furthermore, SHP2 inhibition directly depletes M2 macrophages with pro-tumorigenic properties by inhibiting signaling via the CSF1 receptor (CSF1R) [13].

The activation of alternative receptor tyrosine kinases (RTKs) and downstream signaling cascades, such as the PI3K/AKT pathway, has been implicated in immune checkpoint therapy resistance [104,105,106]. SHP2, as a downstream effector of multiple RTKs, plays a crucial role in these signaling networks. Moreover, SHP2 inhibitors show promise in overcoming resistance to kinase inhibitors and PD-1 blockade, and they can also hinder the activation of compensatory signaling pathways that contribute to immune checkpoint inhibitors (ICIs) resistance. The effectiveness of combining SHP2 inhibitors with other protein inhibitors surpasses single-drug treatments, helping to overcome drug resistance. Encouraging clinical combinations involve SHP2 inhibitors along with immune checkpoint inhibitors like PD-1, as well as inhibitors targeting proteins such as RTK, MEK, ERK, and ALK [14]. By targeting SHP2, it may be possible to attenuate the activation of compensatory pathways and restore sensitivity to immune checkpoint blockade.

## 10. Conclusions

The understanding of SHP2’s molecular structure, functional properties, and signal control has advanced significantly in the last two decades, and its link to associated disorders is crucial in the medical field. In addition to its oncogenic role in cancer cells, SHP2 is involved in various immune cell signaling pathways, making it an attractive target for cancer treatment. Small compounds inhibiting SHP2 and PROTACs are being developed, and 11 allosteric SHP2 inhibitors (such as PD-1, MEK, RTK, ERK, BRAF, and ALK) are undergoing clinical evaluation for cancer treatment. Combining SHP2 inhibitors with other inhibitors has shown promise in halting cancer cell proliferation and reducing drug resistance. However, inhibiting oncogenic SHP2 mutations may require combining inhibitors that work on distinct allosteric locations.

In conclusion, targeting SHP2 represents a promising approach to overcome resistance to immune checkpoint therapy. The modulation of the TME, the enhancement of APC function, and the attenuation of compensatory signaling pathways are potential mechanisms through which SHP2 inhibition can sensitize tumors to ICIs. Further preclinical and clinical investigations are warranted to validate the therapeutic potential of SHP2 inhibitors in combination with immune checkpoint blockade.

## Figures and Tables

**Figure 1 cancers-15-05384-f001:**
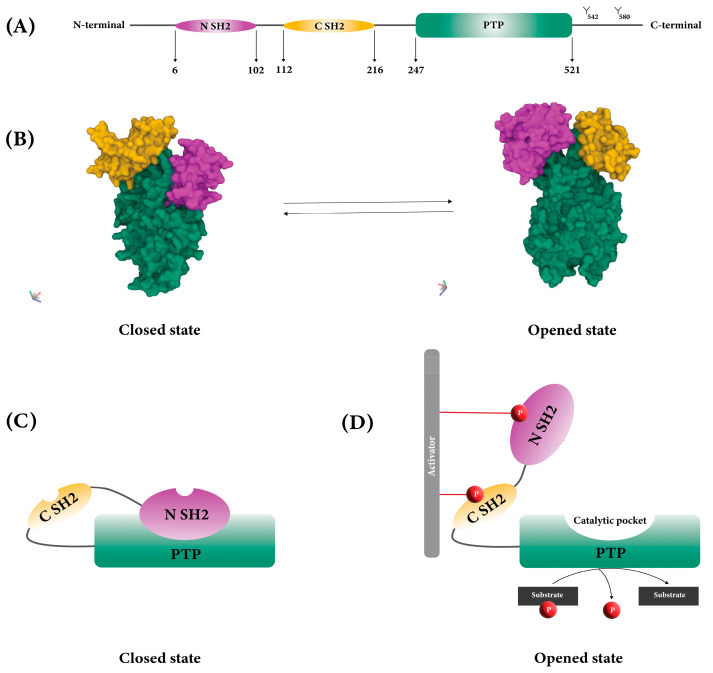
Structure of SHP2. (**A**) Two SH2 domains and PTP domain of SHP2. (**B**) 3D structure of SHP2 in opened (PDB: 6CRF) and closed (PDB: 5EHR) states. (**C**) Auto-inhibition of SHP2 due to the blocking of the PTP catalytic site by N-SH2. (**D**) Exposure of the catalytic pocket due to the conformational change of SHP2.

**Figure 2 cancers-15-05384-f002:**
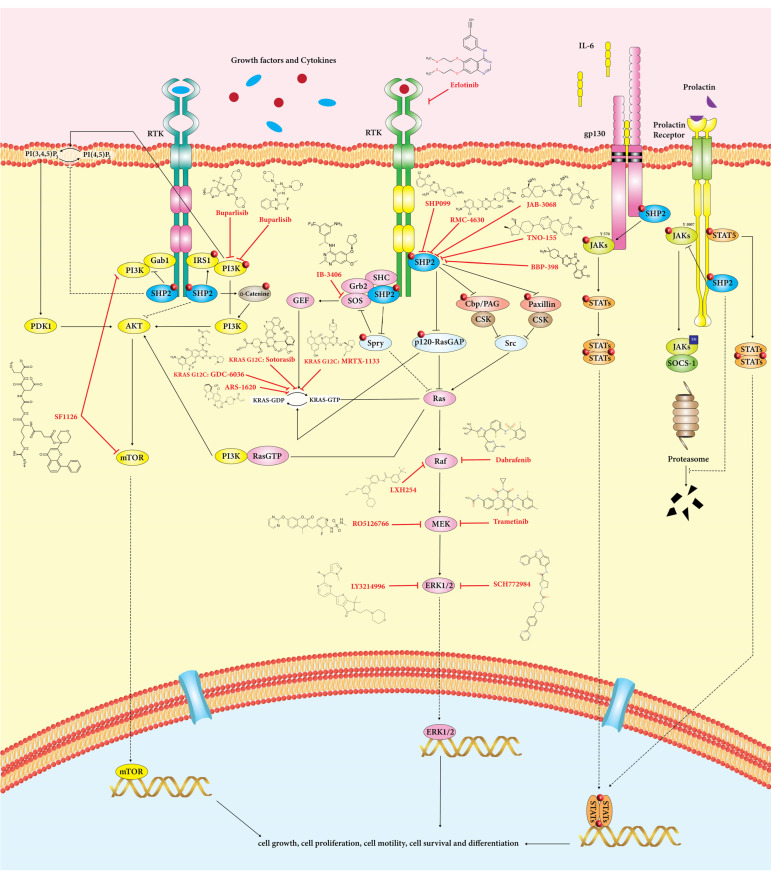
SHP2 is critical in regulating RAS/ERK, PI3K/Akt, and JAK/STAT signaling pathways in cancer cells. These pathways involve cell growth, differentiation, metabolism, and apoptosis. SHP2 positively influences the RAS/ERK signaling pathway. Based on substrate specificity, SHP2 has a dual role in PI3K/Akt and JAK/STAT pathways.

**Table 1 cancers-15-05384-t001:** Other SHP2 allosteric inhibitors.

Name of Compound	Characteristics	Allosteric Site	Cell Line under Study	Target Mutation	Efficacy	Oral Bioavailability	Off-Target Inhibitory Effect	Combination
SHP099 [67,69,70,71,72,73,74]	Suppression of MAPK pathway activity in RTK-driven cancer cell lines and inhibition of malignant development in both in vitro and in vivo tumor models exhibited a synergistic effect on SHP2evoked illnesses or resistant malignancies caused by other mutations.	Tunnel allosteric site	Leukemia cell lines (inflammatory disease)	E69K mutation and leukemia-associated SHP2E69K mutant	Highly effective and selective	Orally accessible	No	BGJ398 (FGFR1 inhibitor) GSK1120212 (MEK inhibitor) Vemurafenib (BRAFV600E inhibitor) BVD523 (ERK inhibitor) Selumetinib (MEK inhibitor)
SHP389 [75,76]	Modulation of MAPK signaling in vivo and poor in vitro permeability	Tunnel allosteric site	Leukemia cell lines (inflammatory disease)	SHP2E76K mutant	Strong hERG selectivity	Limited oral bioavailability	-	-
SHP394 [15]	High lipophilic efficiency, improved potency, and enhanced pharmacokinetic properties SHP93, when administered orally to immunocompromised mice carrying subcutaneously implanted Detroit-562 tumor cells, demonstrated correlated and dose-dependent PK, PD, and effectiveness.	Tunnel allosteric site	Detroit-562 tumor cells, Caco-2 cells, KYSE520 cells	-	Potent, Selective, and Orally Efficacious and reduces tumor volume	Orally active	-	-
SHP836 [70]	In SHP2-WT, the published IC50 value for SHP836 is 5 µm, while the published IC50 value for SHP099 is 70 nm.	Tunnel allosteric site	-	-	SHP836 is much less effective than SHP099.	-	-	-
SHP244 [66,67]	Identified as a weak inhibitor of SHP2 with modest thermal stabilization of the enzyme	Latch allosteric site	JHH-7 and Hep3B cells	SHP2T253M/Q257L double mutant	good selectivity over the catalytic domain	-	-	RMC-4550

**Table 2 cancers-15-05384-t002:** Clinical Trials of SHP2 Inhibitors in Oncology *.

Name of Compound	Company	Stage of Development	Diseases under Study	Combinations
SAR442720 (also known as RMC-4630)	Sanofi (Hongkong, China) (NCT04418661)	Phase 1 and 2	Advanced Solid tumors and KRAS^G12C^ and KRAS mutated solid tumors	LY3214996 (ERK inhibitor) Sotorasib (KRAS^G12C^ inhibitor) Cobimetinib Osimertinib Adagrasib Pembrolizumab (PD-1 inhibitor) AMG510 (KRAS^G12C^ inhibitor)
BBP-398 (Formerly known as IACS-15509)	Navire Pharma Inc., a BridgeBio company (San Francisco, CA, USA) (NCT04528836)	Phase 1 and 2	Solid tumors and NSCLC w/KRAS mutations	Nivolumab (PD-1 inhibitor) in NSCLC
RLY-1971	Hoffmann-La Roche (Basel, Switzerland) (NCT04252339)	Phase 1	Advanced solid tumors	Mono
TNO-155	Novartis (Basel, Switzerland) (NCT04000529) (NCT04330664) (NCT04294160)	Phase 1 and 2	Advanced solid tumors, EGFR-mutated NSCLC	EGF816 (nazartinib) (mutant-selective EGFR inhibitor) Spartalizumab Ribociclib JDQ443 + Tislelizumab (PD-1 inhibitor) Adagrasib Dabrafenib LTT462
ERAS-601	Erasca (San Diego, CA, USA) (NCT04670679) (NCT04959981)	Phase 1/1b	Advanced solid tumors and KRAS mutated NSCLC	Cetuximab Sotorasib
BPI-442096	Betta Pharma (Hangzhou, China) (NCT05369312)	Phase 1	Advanced solid tumors	Mono
ET0038 (development so far in China only)	Etern BioPharma (Shanghai, China) (Chinese) (NCT05354843)	Phase 1	Advanced solid tumors	Mono
HS-10381 (development in China only)	Jiangsu Hansoh Pharmaceutical (Lianyungang, China) (Chinese) (NCT05378178)	Phase 1	Advanced solid tumors	Mono
JAB-3312	Jacobio Pharmaceuticals Co. (Beijing, China), (Chinese) (NCT05288205)	Phase 1 and 2	Advanced solid tumors w/KRAS^G12C^ mutations	JAB-21822 (KRAS^G12C^ inhibitor)
JAB-3068	Jacobio Pharmaceuticals Co., (Chinese) (NCT03565003)	Phase 1 and 2	Advanced solid tumors: NSCLC, squamous esophageal, HNSCC	Mono
HBI-2376	HUYABIO International, LLC. (San Diego, CA, USA) (NCT05163028)	Phase 1	Advanced solid tumors/KRAS or EGFR Mutations	Mono
SH3809	Nanjing Sanhome Pharmaceutical, Co., Ltd. (Nanjing, China) (NCT04843033)	Phase 1	Advanced Solid Tumor	Mono
Sodium stibogluconate	M.D. Anderson Cancer Center/VioQuest Pharmaceuticals (Houston, TX, USA) (NCT00629200)	Phase 1	Advanced Cancer	Mono

* This information is available on clinicaltrials.gov (accessed on 16 June 2023).

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
