# Peer review of "Overcoming Immune Checkpoint Therapy Resistance with SHP2 Inhibition in Cancer and Immune Cells: A Review of the Literature and Novel Combinatorial Approaches"

_cancers, 2023, doi:10.3390/cancers15225384_

Round 1

Reviewer 1 Report (New Reviewer)

Comments and Suggestions for Authors

the whole paper is written in confusion

Comments on the Quality of English Language

the whole paper is written in confusion

Author Response

Thanks for your valuable comments, the authors have re-written the paper with clear mark up and track changes.

Reviewer 2 Report (New Reviewer)

Comments and Suggestions for Authors

a lot of unrelated text that is not well composed into an interesting whole

Comments on the Quality of English Language

ok

Author Response

Thanks for your valuable comments, the authors have re-written the paper with clear mark up and track changes.

Reviewer 3 Report (New Reviewer)

Comments and Suggestions for Authors    'To my idea, the authors have successfully addressed all the comments/questions raised by the reviewers and I reckon this manuscript is now acceptable in the journal.'   'This is a review article, already reviewed by previous reviewers and I was asked to check whether the authors have successfully addressed the comments/questions. To my understanding, this review article is ready for publication.' 

Author Response

Thanks for your valuable comments, the authors have re-written the paper with clear mark up and track changes.

This manuscript is a resubmission of an earlier submission. The following is a list of the peer review reports and author responses from that submission.

Round 1

Reviewer 1 Report

Comments and Suggestions for Authors

In this manuscript, the authors demonstrated the crucial role of SHP2 in immune inhibition and cancer growth, and its potential in overcoming drug resistance as an anti-cancer target of combinatorial therapy. This paper needs to be revised carefully as it is too long and some contents are duplicated. Also, there are some other points need to be addressed before its publication. The followings are my comments and suggestions:

Major points:

1. This paper is too long and it is can be cut down to 5000 words.

2. Some repetitive points can be integrated. As T cells are important components of TME, contents in part5 and part6 can be merged into one part “The immune suppression effects of SHP2 on Tumor Microenvironment”. Part7 and part10part8 and part11 can be integrated. All repetitive contents should be removed.

3. Some substances not so relevant can be deleted or simplified. For example, in Page5, it’s not so necessary to introduce the effects of PTPN11 mutations on NS in detail.

4. We expect more discussion on potential mechanisms of inhibiting SHP2 to overcome immune checkpoint therapy resistance, which should be the central idea of the manuscript.

Minor points:

1.     The second subtitle is too general. It would be best to change the second subtitle to “The Structure and Biological Functions of SHP2”.

2.     Page2, Line4753, there is no need to annotate abbreviation “SH2” more than once.

3.     Page3, Line 119, Page5, Line 158160 and many other places in the manuscript, please change all “Shp2” to “SHP2” to unify writing.

4.     Page3, Line 134, “promoted” should be “are promoted”.

Comments on the Quality of English Language

 Minor editing of English language required.

Reviewer 2 Report

Comments and Suggestions for Authors

The review article entitled ‘Overcoming Immune Checkpoint Therapy Resistance with SHP2 Inhibition: A Review of the Literature and Novel Combinatorial Approaches ‘ was well received. The article focuses mainly on SHP2 inhibition in cancer. So, the title could be misleading and it should be specified in the title

In the introduction section, please explain immune check point inhibitory therapy resistance mechanisms. The article should first explain immune checkpoint therapy resistance followed how can SHP2 inhibition overcome these resistance mechanisms.

There are multiple formatting issues in the paper for example the after a sentence is finished references are added without spaces. There must be a space. Please rectify

Comments on the Quality of English Language

English language is fine.  

Round 2

Reviewer 2 Report

Comments and Suggestions for Authors

The authors have not responded to the comments properly and failed to satisfy the raised concerns. Immune checkpoints are certain receptors that are displayed on the surface of Immune cells (NK and T most importantly). These receptors get activated by binding their ligands on the cancer cell surface and initiate negative signaling in effector immune cells most of the time via SHP2 activation (in immune cells).  

I am obliged to follow the Title of the manuscript. According to the title, the manuscript should be able to deliver following information which it failed to do.    

What is Immune check point therapy?

Immune check point therapy resistance mechanisms?

SHP2 role in check point therapy resistance?

SHP2 inhibitors

The authors must clarify by SHP2 inhibition… do they mean SHP2 inhibition in cancer or SHP2 inhibition in Immune cells or Both. Please clarify.

If they mean SHP2 inhibition both in Cancer and Immune cells, they must discuss SHP2 signaling in Cancer and Immune cells separately before discussing the impact of inhibition in both.

This paper contains tons of materials they are not organized well. The authors can squeeze the size of paper and present information in a bit specific and organized manner.

Comments on the Quality of English Language

Fine

Author Response

We are grateful for the constructive feedback provided by the reviewer. We believe that the revisions made have addressed the concerns raised and have improved the overall quality of our manuscript. We hope the changes meet the reviewer's expectations and look forward to further feedback.

Please see attached response to reviewer 2 and resubmitted manuscript with highlights.
